# Utilization of Healthcare Services in Patients with Chronic Diseases under 18 Years Old: Differences and Contributing Factors

**DOI:** 10.3390/jpm14090956

**Published:** 2024-09-09

**Authors:** Jaime Barrio-Cortes, Beatriz Benito-Sánchez, Raquel Sánchez-Ruano, César Alfonso García-Hernández, María Teresa Beca-Martínez, María Martínez-Cuevas, Almudena Castaño-Reguillo, Cristina Muñoz-Lagos

**Affiliations:** 1Foundation for Biosanitary Research and Innovation in Primary Care (FIIBAP), 28003 Madrid, Spain; jaime.barrio@salud.madrid.org (J.B.-C.); rsruano@salud.madrid.org (R.S.-R.); 2HM Faculty of Health Sciences, Camilo José Cela University, 28692 Madrid, Spain; mariateresa.beca@ucjc.edu; 3Gregorio Marañón Health Research Institute (IiSGM), 28009 Madrid, Spain; 4Research Network on Chronicity, Primary Care and Prevention and Health Promotion (RICAPPS), Spain; 5Ciudad Jardín Health Centre, Primary Care Assistance Management, 28002 Madrid, Spain; 6Segre Health Centre, Primary Care Assistance Management, 28002 Madrid, Spain; cesaralfonso.garcia@salud.madrid.org; 7Fuencarral Health Centre, Primary Care Assistance Management, 28034 Madrid, Spain; mmartinezcuevas@salud.madrid.org; 8Los Ángeles Health Centre, Primary Care Assistance Management, 28041 Madrid, Spain

**Keywords:** pediatrics, chronic disease, medical complexity, health services utilization, primary care, hospital care, care management, patient-centered care

## Abstract

Pediatric patients with chronic conditions frequently have unmet care needs, make extensive use of healthcare services, and often encounter fragmented, non-centered care. This study aimed to analyze the differences in the utilization of primary care (PC) and hospital care (HC) services by these patients according to sex, age, and complexity and to identify associated factors. A cross-sectional study was conducted in a basic health area of Madrid, including patients under 18 years. Among these patients, 15.7% had ≥1 chronic disease, 54.1% were male, the average age was 9.5 years, 3.5% had complexity, and 11.3% had multimorbidity. The mean number of contacts/year with the healthcare system was 9.1, including 8.3 contacts/year with PC (4.9 with pediatricians and 1.9 with nurses) and 0.8 contacts/year with HC (0.8 in external consultations and 0.01 hospitalizations). The factors associated with PC utilization were complexity; female sex; European origin; and diseases like asthma, epilepsy, stroke, recurrent urinary infection, attention deficit hyperactivity disorder, and anxiety, while older age was negatively associated. Thyroid disorders were significantly associated with HC utilization. These findings could help guide the design of optimized pediatric patient-centered care approaches to coordinate care across healthcare services and reduce high healthcare utilization, therefore improving the healthcare outcomes and quality of life for these patients.

## 1. Introduction

The prevalence of chronic diseases in the pediatric population has increased dramatically in recent years, mainly due to improvements in pediatric care that have led to higher survival rates, together with diagnostic advances that have made it possible to detect diseases that previously would have gone unnoticed [1].

Definitions regarding chronic diseases in pediatrics are very heterogeneous, making it difficult to accurately and reproducibly estimate their prevalence [2]. Most of the definitions combine different criteria of chronicity, including the duration of the disease and its consequences according to functional limitations and/or care needs [3]. The estimated prevalence of chronic diseases in the pediatric population varies widely, from less than 10% to almost 50% [2]. This prevalence also depends on the geographical area studied, the specific subpopulation analyzed, and the source of information [4].

Chronic diseases in pediatric patients require frequent visits to healthcare services due to ongoing management needs, care involving multiple specialists, regular medication adjustments, continuous monitoring, patient education, and special preventive care measures [5,6]. Given the increase in chronic diseases in the pediatric population, healthcare systems must also evolve to provide appropriate treatment and care to these patients since they often encounter the fragmentation of existing healthcare systems [7]. Novel integrated patient-centered care approaches should be developed to organize and coordinate care across multiple systems in a continuous and efficient way, thus managing to meet the specific needs of each patient [8]. To guide the design of novel optimized healthcare models, it is fundamental to first understand the characteristics of the pediatric population with chronic diseases and their use of healthcare services. Understanding healthcare utilization in pediatric patients with chronic conditions is especially crucial for optimizing care and reducing costs. However, evidence on this topic is limited, and more research is highly needed [9,10].

In recent decades, various systems have been developed to classify morbidity, such as the Clinical Risk Group (CRG) [11], the Adjusted Clinical Group (ACG) [12,13], and the Adjusted Morbidity Group (AMG), the latter created in Spain [14]. These highly useful clinical and management tools stratify the population according to their level of risk and make it possible to predict their use of health services. However, in pediatric patients, grouping by medical complexity could be more appropriate than by risk level [15].

Pediatric patients with complexity are characterized by one or more severe conditions and/or chronic diseases associated with higher frailty, more functional limitations, and a greater need for care and use of services [16]. Despite the low prevalence of cases with complexity, estimated to be between 0.4% and 6% of the pediatric population [17], almost one-third of the expenses in pediatric care are due to these patients with complexity [18]. Many of them are commonly referred to as ‘super-users’, defined as ‘beneficiaries with complex, unaddressed health issues and a history of frequent encounters with healthcare providers’, including an average of at least three hospital visits per year [19]. 

However, it is important to note that clinical complexity in pediatric patients is not solely influenced by medical conditions. Social determinants of health, such as family income, parental education, and access to healthcare services, also play a significant role in the overall health and care utilization of these children. These factors can exacerbate the challenges faced by patients with complex conditions, leading to increased healthcare utilization and poorer outcomes if not adequately addressed [20].

This situation results in the needs of chronic pediatric patients being frequently unmet and uncoordinated, and therefore, they make substantial utilization of healthcare services, resulting in high expenses for the healthcare system [5]. This significant issue highlights the necessity of an exhaustive analysis and understanding of their healthcare needs and utilization to guide the design of optimized and cost-effective, patient-centered care strategies. These strategies should involve the redistribution of resources and healthcare professionals to meet the individual needs of these patients and improve their quality of life [8]. Additionally, the stratification of this population according to sociodemographic and clinical characteristics could be very useful to tackle the different needs of the groups identified since not all chronic pediatric patients require the same management approaches, and they do not make the same use of services [21,22]. For this purpose, the aim of this study was to analyze the differences in the characteristics and use of primary care (PC) and hospital care (HC) services among pediatric patients with chronic diseases according to sex, age, and complexity, as well as the factors associated with their use.

## 2. Materials and Methods

### 2.1. Study Design

A descriptive cross-sectional study with an analytical approach was conducted. 

### 2.2. Study Setting and Population

This study was developed in a basic health area and its referral hospital in the district of Chamartín located in Madrid (Spain) (Appendix A). In Spain, healthcare services are distributed following a region-based organization into basic health areas that attend to demographic and geographic criteria, with each health area constituting the territorial framework where PC centers and a general hospital to which patients are referred are based [23]. Chamartín is a district in the northern area of Madrid city with a population of 143,424 inhabitants and a mean age of 45 years, with 55% women and 8.9% foreigners. According to Medea Index, Chamartín corresponds to the lowest degree of socioeconomic deprivation [24,25]. The district is characterized by a mix of residential, commercial, and cultural areas featuring mid-to-high-income housing, shopping centers, and parks. Well-connected by major roadways and public transport, it also boasts several healthcare facilities, including clinics and hospitals, that provide comprehensive medical services to its diverse population. The basic health area and its referral hospital studied had 18,107 registered patients, of whom 2691 were under 18 years of age, as of 30 June 2015.

### 2.3. Study Sample

To define the study sample, patients 18 years or older were excluded. And among the 2691 patients <18 years of age, only those identified as chronic by the AMG tool were included in the study. AMG is integrated into the PC digital medical records of the Community of Madrid and identifies as chronic any individual who suffers from at least one of the chronic diseases described in Appendix A, according to the Strategy for the Care of Chronic Patients of the Community of Madrid [26].

### 2.4. Source of Information and Data Collection

The digital medical records of the Community of Madrid served as the source of patient information. These data were obtained from the Information Health Systems Department of Madrid PC and HC after having received ethical approval. The different anonymized PC and HC databases received were then linked and combined into a single database to integrate the information effectively and facilitate analysis.

### 2.5. Variables Analyzed

The AMG grouper stratifies the population according to its morbidity and complexity. This tool calculates a numerical complexity index while considering the healthcare needs of users based on variables such as mortality, risk of admission, PC visits, and prescription. Subsequently, the calculated complexity index allows patients to be grouped into different risk levels: no risk, low risk, medium risk, or high risk [14].

The following sociodemographic variables were extracted from the PC digital medical records as of 30 June 2015: age (quantitative), sex (qualitative), and country of birth (qualitative); as well as clinical care variables, including the complexity index according to the AMG (quantitative value that scores the morbidity and care needs of the patient) [14]. The level of complexity was determined according to the AMG (qualitative with ‘no complexity’ for low-risk patients and ‘complex’ for medium- or high-risk patients) [14,27], the number of chronic diseases (quantitative), presence of multimorbidity (qualitative defined as 2 or more chronic conditions), and the suffering from the chronic diseases and associated severe acute comorbidities (qualitative) identified by the AMG tool of the digital medical records of the Community of Madrid (Appendix A). The variables related to the use of PC services were quantitative and were extracted from the PC digital medical records: number of annual contacts, type of contact (health, laboratory, and administrative), form (face-to-face, telephone, or home visit), and professional (pediatrician, nurse, and family doctor). HC services were extracted from HC digital clinical records and were number of annual contacts and place (external consultations, hospitalization, and day hospital visits). Both PC and HC variables were extracted for the period of one year, between 30 June 2015 to 30 June 2016.

### 2.6. Statistical Analysis

For descriptive statistics (univariate analysis), qualitative variables were expressed as frequencies and percentages, while quantitative variables were expressed as means and their respective standard deviations as measures of dispersion. The normality of the qualitative variables was assessed using the Shapiro–Wilk test. In the bivariate analysis, chi-square tests or Fisher’s exact tests were used for qualitative variables, the Mann–Whitney U test was used for qualitative and quantitative variables, and the Kruskal–Wallis test was used for polytomous and quantitative variables. The results of the multiple comparisons were adjusted using the Bonferroni correction. The factors associated with the use of services were analyzed using Poisson regression. Statistically significant results were those with a value of *p* < 0.05. The statistical analysis was performed with IBM SPSS Statistics version 25 software (IBM Corp., Armonk, NY, USA).

## 3. Results

Of the 2691 patients <18 years of age registered in the basic health area, 423 (15.7%) suffered from at least one chronic disease. Of these, 54.1% were male, 74.7% were Spanish, and their average age was 9.5 years. Their average complexity index was 3.6, and 3.5% of them were identified as having complexity. Their mean number of chronic diseases was 1.1, and 11.3% presented with multimorbidity. There were statistically significant differences between PC users and non-PC users: the mean age was lower in PC users, with the most prevalent age group being 0–4 years; there were more non-national users among PC users; and the complexity index was higher in PC users compared to non-PC users. However, no other statistically significant differences were found between HC users and non-HC users (Table 1).

In total, 87.7% of the chronic pediatric patients were contacted with PC in the year studied. In this cohort, 54.4% were male, with an average age of 9.3 years for both sexes and a higher complexity index in males (3.8 vs. 3.6). PC users under 4 years of age had higher complexity, with 6.0 as the average index and 52.1% of them being girls. A total of 3.8% of the PC users presented with complexity, of whom 71.4% were male, 3.7 years was the average age, and 12.2 was the average complexity index (Appendix A).

On the other hand, 28.1% were HC users, of whom 52.9% were men, their average age was higher than that of females (9.7 vs. 8.6 years), and their average complexity index was lower (3.5 vs. 3.7). HC users aged less than 4 years old presented with higher complexity, with an average complexity of 6.6, with 61.9% being girls. A total of 2.5% of the HC users were identified with complexity, of whom 66.7% were male, the average age was 2.3 years, and the mean complexity index was 14.6 (Appendix A).

The most prevalent comorbidities in pediatric patients were asthma (38.8%), attention deficit hyperactivity disorder (ADHD) (11.3%), anemia (10.6%), pneumonia (9.9%), obesity (8.7%), thyroid disorders (6.4%), hyperlipidemia (5.4%), anxiety (5.4%), and epilepsy (4.7%). Statistical significance differences were found in relation to certain comorbidities: pneumonia and valvular disease were more common in primary care (PC) users compared to non-PC users, while thyroid disorders and paralysis were more prevalent in HC users compared to non-HC users (Table 2). 

In PC users, for both sexes, the most prevalent comorbidity was also asthma (39.4%), followed by pneumonia (11.8%) in girls and ADHD (12.4%) in boys. PC users under 4 years of age frequently presented with asthma and pneumonia; those between 5 and 9 years old frequently had asthma followed by obesity; those between 10 and 14 years old had pneumonia and ADHD; and those older than 15 years old had asthma and anxiety. The percentage of PC users with complexity who presented with diseases such as asthma, epilepsy, anemia, pneumonia, and thyroid disorder was higher than that of patients without complexity (Appendix A).

Among the HC users, the most frequent comorbidities were asthma (37.8%), ADHD (11.8%), and thyroid disorders (10.9%). In boys, the most frequent comorbidities were asthma, ADHD, and anemia, and in girls, the most frequent comorbidities were asthma, thyroid disorder, and pneumonia. At <4 years of age and between 5 and 9 years of age, asthma and pneumonia were common; from 10 to 14 years of age asthma, ADHD and anemia were common; and over 15 years of age, asthma, anxiety, and ADHD were common. HC users with complexity presented with asthma, epilepsy, neoplasia, or paralysis (Appendix A).

Regarding the use of services, 91.3% of pediatric patients with chronic diseases contacted the health system during the period studied, registering an average of 9.1 contacts/year (Table 3).

The mean number of contacts/year with PC was 8.3, with the most common type being a health consultation (percentage of patients with ≥1 visit [n] = 87.7%; mean [M] = 7.9 contacts/year), and the most common form being face-to-face visits (*n* = 87.5%; M = 8.2). The most consulted professional was the pediatrician (*n* = 72.1%; M = 6.3), followed by the nurse (*n* = 67.8%; M = 1.9) and the family doctor (*n* = 19.1%; M = 0.7) (Table 4).

Regarding the use of HC, an average of 0.8 annual visits were recorded; these visits were distributed among external consultations (*n* = 27.9%; M = 0.8), hospitalizations (*n* = 1.2%; M = 0.01), and hospital day visits (*n* = 0.5%; M = 0.02) (Table 5). 

The female sex registered a higher annual utilization of healthcare services (9.9 vs. 8.5 for men), 9.1 with PC (7.7 for men) and 0.8 with HC (0.9 for men). The age group with the most health visits were those under 4 years of age, with an average of 13.9 contacts/year; patients between 5 and 9 years registered 9.2 contacts/year; and those older than 10 years registered 7.4 visits/year. Patients with complexity made an average of 22.7 visits/year, 22.5 with PC and 0.2 with HC, and those without complexity made 8.6 visits/year, 7.8 with PC and 0.9 with HC (Appendix A).

The variables that explained a higher use of the PC with statistically significant differences included complexity, female sex, and European origin, along with the diseases pneumonia, asthma, epilepsy, cerebrovascular accident, recurrent urinary tract infection, ADHD, hyperlipidemia, thyroid disorder, and anxiety. In contrast, older age and arthritis were associated with a lower use of PC. Regarding HC, only patients with thyroid disorders showed a statistically significant higher use of these services, while those with hyperlipidemia and recurrent urinary infections showed a negative correlation (Table 6). 

## 4. Discussion

Unmet care needs in chronic pediatric patients and their extensive healthcare utilization emphasize the need for novel patient-centered care approaches, which must be designed according to the specific needs and healthcare utilization of these patients. However, there is very limited evidence of their characteristics and the use of PC and HC services. The present study contributes to addressing this challenge by describing the characteristics and use of health services by pediatric patients with chronic conditions, resulting in the first study to segment the analysis into PC and HC users, as well as by sex, age group, and complexity.

Of the pediatric population studied, 15.7% had chronic diseases, which is within the range reported in an extensive systematic review on chronic health conditions in childhood [2]. Slightly more than half of the patients were male, as were the users of PC and HC, which is in line with the findings of other studies [4,5,28,29,30]. The mean age of the population was 9.5 years, which is very close to that described by Hoefgen et al. [5], although higher than that observed by other authors [31]. These differences in age between studies can be explained by the definition used and by the geographical area, population, and clinical environment considered. The most common diseases observed in our chronic pediatric population were among the classes of diseases described in the literature as being responsible for most of the growth of chronic conditions among pediatric patients globally, these being asthma or other respiratory conditions, mental health conditions, obesity, and neurodevelopmental disorders [1].

A total of 3.5% of chronic pediatric patients had complexity, fitting in the estimate between 0.4% and 6% observed in the literature [17]. In line with the findings of previous studies, a higher proportion of patients with complexity were men [18,28,29], and the average age was reduced to 3.7 years [18]; however, other publications confirm higher means or a higher mean age for patients with complexity than for patients without complexity [5,28,31]. The decrease in the age of patients with complexity may be due to their complex condition, which increases early mortality rates and reduces their life expectancy [32]. The mean age of our patients with complexity was consistent with the data recorded in Spain during our study [33]. In patients with complexity, certain pathologies, such as asthma, epilepsy, and neoplasia, were significantly more frequent, probably due to their more severe nature and/or because of their coexistence with other more complex health problems, according to the observations of other analyses [28]. In line with the literature [29], the percentage of patients with multimorbidity was significantly higher among those with complexity compared to those without.

### 4.1. Use of Services

The average number of contacts with the health system of the chronic pediatric population exceeded nine contacts/year, doubling the annual number of visits of the total pediatric population in the Community of Madrid and at the national level in the year of our study [34,35].

Complex patients registered an average of close to 23 healthcare contacts/year, resulting in a remarkably high number supporting the fact that the level of healthcare use is significantly higher for patients with complexity than for the chronic pediatric population and much higher than for the general pediatric population [18]. 

Despite this, complexity is not the only factor that has a big impact on pediatric health outcomes. Socioeconomic factors also influence a child’s well-being and development by affecting access to necessary resources, the quality of the environment where they grow up, and the ability to receive timely healthcare. Addressing these factors is essential to effectively tackle health disparities and promote equity in health outcomes. Research indicates that children from lower-income families or those living in deprived areas are at higher risk for a range of adverse health outcomes, including developmental delays, chronic health conditions, and reduced access to healthcare services [20]. The high average number of healthcare contacts among pediatric patients in our study may have been potentially influenced by their urban location and lower levels of socioeconomic deprivation.

This extremely high use of healthcare services by both the chronic pediatric population and those complex patients, most of whom are super users, highlights the need to design and implement novel optimized management approaches to provide better care for pediatric patients to meet their specific needs and to ensure a better quality of life while also addressing the necessity to reduce the frequency of visits. 

Regarding the possible impact of the COVID-19 pandemic in this framework, a study in the USA in 2022 found that the pandemic led to a significant decrease in the use of healthcare resources for pediatric patients, with a more pronounced reduction among children with chronic conditions compared to healthy children. This decline persisted during the first twelve months of the pandemic for chronic patients, while visits for healthy children gradually returned to pre-pandemic levels. This decrease was partly due to families’ concerns about exposing chronic patients to potential infections. Mental health-related visits saw a smaller drop and even a slight increase after the first pandemic year. Additionally, antibiotic prescriptions fell, but there was an uptick in prescriptions in the months following the pandemic year, whereas antidepressant prescriptions remained unchanged [36]. In contrast, another study in Norway in 2022 concluded that COVID-19 had a limited impact on healthcare services for children and adolescents [37].

Vaccinations also influence healthcare utilization among all children by increasing routine healthcare visits. These visits, often conducted by nurses, provide essential opportunities not only for administering necessary immunizations but also for engaging in health promotion and chronic disease management. This proactive approach contributes to improved health outcomes and more efficient use of healthcare resources. However, children with chronic illnesses typically have lower vaccination rates compared to their healthy peers, placing them at greater risk if they contract a preventable disease [38].

### 4.2. Use of PC Services

The number of users, as well as the average number of contacts with PC, was much higher than that with HC, supporting the care models developed in recent years aimed at this profile of patients with special needs, defending PC as the most appropriate health level for their care and for providing them with continuity of care [39]. Among the few studies that describe PC frequentation by these patients, Cohen et al. identified a median of 12 visits in 2 years by complex patients in Canada [17], much lower than the median of 19 annual visits by complex patients identified in our study, revealing that, although the health systems of both countries are universal and focused on primary care [35,40], in Spain there is an urgent need to reorganize the PC resources in order to provide better and optimized care for these patients to prevent complications of their diseases, thus consequently reducing their number of consultations with PC and the burden on PC professionals.

The most frequently consulted primary care professional was the pediatrician, as they are the first point of contact with the health system for pediatric patients. Pediatricians handle over 90% of pediatric demands, oversee diagnosis and monitoring, and coordinate with other health professionals [39]. Unlike the pediatric care models in some European countries that focus on family doctors, the Spanish model relies on pediatricians. This approach has proven to be the most cost-efficient, yielding better results and receiving widespread acceptance from the population [41]. For example, in comparison to a pediatric care model primarily focused on family doctors, which reported that 90.5% of patients under 18 with long-term chronic diseases visited either a general practitioner or a pediatrician at least once in the studied year [30], our findings showed that only 83.5% of patients required attention from either of these professionals. 

Despite the significantly higher number of nurse visits compared to other countries’ healthcare systems [30], the Strategy for the Care of Chronic Patients and proposals for improving pediatric care in the Community of Madrid suggest that nurses should play a more prominent role, sharing the care burden with pediatricians [26,41].

The female sex was superior to the male sex in terms of PC utilization, in line with the findings of previous studies [42,43]. This suggests that there are differences between the sexes regarding the manifestation, epidemiology, and pathophysiology of many diseases and in the form of contact with the health system. European patients made more use of PC, probably due to poor communication with professionals due to linguistic differences [44]. This situation was not observed in patients from outside of Europe since the majority came from a Spanish-speaking country. In contrast, older age was negatively correlated with the use of PC since younger patients tend to have more care needs [18,32,33,42,43].

Greater use of PC was also associated with higher complexity, confirming the findings described in the literature [18]. The diseases most strongly associated with PC use were pneumonia and asthma since a large percentage of PC pediatric visits worldwide are related to respiratory diseases, especially during the first years of life [45]. Other comorbidities related to the use of PC were epilepsy and cerebrovascular accidents since they need to be periodically controlled by health professionals to avoid the development of severe, irreversible problems [46]. Recurrent urinary tract infection was also associated with higher attendance at PC since its diagnosis in the pediatric population is complicated, requiring several medical consultations until it is correctly diagnosed and can lead to medical complications [47]. Patients with ADHD, anxiety, hyperlipidemia, and thyroid disorder also registered more visits to PC, probably because of the periodic monitoring of their condition and the continuous regulation of medication [48,49].

### 4.3. Use of HC Services

The frequency of HC use by the chronic pediatric population was lower than that observed in other studies [4,5,18] but consistent with the care models identified as more appropriate, with a preference for PC fulfilling a preventive role [39]. Most of the contacts with HC occurred in external consultations and day hospitals and not at admission, revealing adequate follow-up and treatment of these patients by specialists and avoiding the development of medical complications that involved hospitalization.

Compared to previous studies, only 1.2% of chronic patients in our study were hospitalized, which is notably lower than the 10.1% of chronic patients admitted over two years described by Hoefgen et al. [5]. Additionally, the mean number of hospitalizations in our study was near the range between 0.03 and 0.22 of inpatient visits reported by Kuo et al. [50]. However, our findings align with both studies in showing that patients with complexity experienced a higher number of hospitalizations [5,50]. Only 4.2% of HC users were hospitalized during the follow-up year, which is lower than the percentage reported for the pediatric population in other hospital records [4,18,32].

There was a slightly higher use of HC by males than females, which is in line with the findings of previous studies [4,18,32]. Patients aged 5 to 9 years old were those who made a slightly higher use of HC, while in other studies, those 4 years of age used HC more [4,18,32]. This disparity could be due to the lower number of HC users in our study.

Unlike previous studies [4,32], we did not find significant factors associated with the use of HC, except for thyroid disorders. It is likely that an increased number of patients using HC in the study might reveal additional significant factors associated with HC use in the pediatric population.

Our work differs from the approach used by Mangione-Smith and her Pediatric Medical Complexity Algorithm (PMCA), which identifies a small proportion of children with complex chronic diseases from Medicaid claims and hospital discharge data, demonstrating good sensitivity and specificity [51]. In contrast, the AMG includes patients detected from the PC setting and includes both complex and non-complex cases, offering more comprehensive evidence of a tool not well-studied or widely used for children in Spain. While studies using PMCA are insightful and useful for understanding complex conditions [52], our study provides a more comprehensive analysis by including patients mainly from the PC electronic clinical record and identifying both complex and non-complex cases. We emphasize the utility and potential of the AMG in the pediatric population, which is not well-studied or widely used for children in Spain.

### 4.4. Limitations and Strengths

Chronic patients were identified from the total pediatric population by means of the AMG tool, which considered the chronic diseases described in the Strategy for the Care of Chronic Patients of the Community of Madrid (Appendix A) [8,26]. It is possible that some specific chronic pediatric diseases were excluded, resulting in an underestimation of the prevalence [15]. However, this limitation is applicable to all studies of pediatric populations since there is currently no agreed definition upon which pathologies are considered chronic pediatric diseases [2].

The number of pediatric patients with chronic diseases and complexity analyzed was not very high. However, our findings are similar and comparable to those stated in other studies with larger samples. Some variables that could be of interest to analyze, such as parental education and income, could not be obtained from our data source due to the lack of this information in most of the electronic clinical records of primary care. Nevertheless, these factors are often underrepresented or inconsistently documented in electronic health records, making it challenging to study their full impact. This lack of data makes it a challenge to identify at-risk populations and develop targeted interventions.

Despite this, our study overcomes the limitations of those studies based on surveys since it has been performed with Real World Data by accessing a large volume of information in real clinical practice conditions, which include a wide range of variables.

Even though only one basic health area was studied, the results described are representative of the chronic pediatric population in the city of Madrid and could be extrapolated to other urban areas in Spain, as well as to other cities in countries with similar characteristics. This is because the studied area provides health coverage to a highly diverse population, including patients of different nationalities and a wide range of pathologies. Furthermore, the results align with the findings of studies carried out in other regions of the world [37,53].

Although the data analyzed were obtained a few years ago, the results described are consistent with those of recent studies. While the use of services may have changed, the numbers of visits to the health system described in the Madrid Health Service report of the year of our study [34] are very similar to those in the last published annual report [54]. Thus, the results regarding the use of services described in our study represent the current reality and can be used to guide the efficient distribution of healthcare resources. Additionally, we could not consider the day and time of utilization, which could influence healthcare service use.

Regarding the use of HC, the AMG was developed in the field of PC, so it may have a lower explanatory power in HC [14]. Despite this, the AMG allows us to analyze the differences in use between PC and HC, providing information of great relevance at the clinical and management levels. It should also be noted that we could not determine visits to the emergency room, as this information was not available at the referral hospital. Finally, this study only considered the use of public healthcare services due to the lack of linkage between public and private electronic clinical records. This may lead to an underestimation of healthcare utilization, as some patients might have double insurance coverage or prefer private HC over public ones. Despite this, PC, public healthcare services, mainly especially PC, serve nearly the entire pediatric population. The widespread reach of these services, particularly through vaccination and medication programs, likely influences overall PC utilization [55].

Despite these limitations, our research fills a gap in the literature by quantifying the utilization of health services in PC and HC among children according to the AMG, differentiating those with complex conditions from non-complex children. This aspect, as we have emphasized, is scarce in existing studies and holds significant value for health planning purposes related to children with chronic diseases. This study addresses critical aspects of patient-centered care and aligns with the goals of defining proactive programs and ensuring evidence-based practices.

### 4.5. Implications

This study underscores several important considerations for enhancing healthcare delivery in pediatric populations with chronic diseases that decision makers could take into consideration. Addressing these areas can improve patient outcomes and optimize resource utilization. To reduce the number of visits, it is essential to develop and strengthen preventive care and patient-centered care management programs. These initiatives are particularly important for managing complex and severe conditions, as they can help prevent complications and health issues from escalating, thereby reducing the need for frequent visits. Additionally, improving care coordination between primary care providers and hospital specialists can minimize unnecessary visits by ensuring patients receive timely and appropriate care promptly. Expanding the role of digital health (e.g., telemedicine, remote monitoring, and apps) offers a convenient alternative to in-person visits, improving access to care, overcoming geographical barriers, and ensuring continuity of care. It facilitates timely interventions and better coordination among healthcare providers. Additionally, digital health provides educational resources and easier communication with healthcare providers, but addressing technology access and privacy concerns is essential. Together, these strategies aim to improve healthcare efficiency and patient outcomes, leading to a more effective use of resources.

## 5. Conclusions

The findings are significant because the evidence on the characteristics and healthcare utilization of pediatric chronic patients is very limited in the literature. Within the studied pediatric population, a notable proportion had chronic diseases and exhibited remarkably high service usage, especially among those with complexity. This highlights the need for novel, optimized care management approaches. Key factors associated with primary care (PC) use include complexity, younger age, female sex, and certain prevalent conditions requiring more care.

It is necessary to improve stratification tools for the pediatric population by incorporating additional chronic comorbidities; the use of PC and HC services; and addressing mental, psychological, and social well-being issues. Additionally, more detailed analyses of these patients’ clinical and socioeconomic characteristics and healthcare use are necessary to better identify their needs. The results of our study could help guide researchers, healthcare professionals, and policymakers in creating novel, optimized, patient-centered strategies. These strategies should involve the reorganization of current health systems to achieve better management of chronic pediatric patients, ultimately improving healthcare outcomes and quality of life for these patients and their families.

## Figures and Tables

**Table 1 jpm-14-00956-t001:** Sociodemographic and clinical care characteristics of pediatric patients with chronic diseases and according to their use of services.

Variables	Total 423 (100)	95% CI	PC Users 371 (87.7)	95% CI	*p*	HC Users 119 (28.1)	95% CI	*p*
Sociodemographic								
Male sex, *n* (%)	229 (54.1)	49.4–58.9	202 (54.4)	49.4–59.5	0.73	63 (52.9)	43.8–62.0	0.76
Age M (SD)	9.5 (4.7)	9.1–10.0	9.3 (3.7)	8.8–9.8	<0.01 *	9.2 (4.6)	8.3–10.0	0.35
Age Group 0–4 years, *n* (%)	76 (18.0)	14.3–21.6	73 (19.7)	15.6–23.7	0.01 *	21 (17.6)	10.7–24.6	0.92
5–9 years, *n* (%)	129 (30.5)	26.1–34.9	114 (30.7)	26.0–35.4	0.78	43 (36.1)	27.4–44.9	0.12
10–14 years, *n* (%)	138 (32.6)	28.1–37.1	116 (31.3)	26.5–36.0	0.11	32 (26.9)	18.8–35.0	0.12
15–17 years, *n* (%)	80 (18.9)	15.2–22.7	68 (18.3)	14.4–22.3	0.41	23 (19.3)	12.1–26.5	0.89
Origin Spain, *n* (%)	316 (74.7)	70.6–78.9	264 (71.2)	66.5–75.8	<0.01 *	87 (73.1)	65.0–81.2	0.64
Rest of Europe, *n* (%)	21 (5.0)	2.9–7.0	21 (5.7)	3.3–8.0	0.08	7 (5.9)	1.6–10.2	0.59
Rest of the world, *n* (%)	86 (20.3)	16.5–24.2	86 (23.2)	18.9–27.5	<0.01 *	25 (21.0)	13.6–28.4	0.83
Clinical								
Complexity index, M (SD)	3.6 (2.6)	3.3–3.8	3.7 (2.6)	3.5–4.0	<0.01 *	3.6 (2.8)	3.1–4.1	0.98
With complexity, *n* (%)	15 (3.5)	1.8–5.3	14 (3.8)	1.8–5.7	0.50	3 (2.5)	0.0–5.4	0.48
Chronic diseases ^a^, M (SD)	1.1 (0.4)	1.1–1.2	1.1 (0.4)	1.1–1.2	0.80	1.1 (0.3)	1.1–1.2	0.53
Multimorbidity, *n* (%))	48 (11.3)	8.3–14.4	42 (11.3)	8.1–14.6	0.96	13 (10.9)	5.2–16.6	0.86

^a^ M (SD): Mean (standard deviation). * *p*-value < 0.05. CI: Confidence Interval; HC: hospital care; PC: primary care.

**Table 2 jpm-14-00956-t002:** Most prevalent comorbidities in pediatric patients with chronic diseases and according to their use of services.

Variables *n* (%)	Total 423 (100)	95% CI	PC Users 371 (87.7)	95% CI	*p*	HC Users 119 (28.1)	95% CI	*p*
Asthma	164 (38.8)	34.1–43.3	146 (39.4)	34.4–44.3	0.51	45 (37.8)	29.0–46.7	0.80
ADHD	48 (11.3)	8.3–14.4	40 (10.8)	7.6–14.0	0.33	14 (11.8)	5.9–17.6	0.87
Anemia	45 (10.6)	7.7–13.6	38 (10.2)	7.1–13.3	0.48	11 (9.2)	4.0–14.5	0.56
Pneumonia	42 (9.9)	7.1–12.8	41 (11.1)	7.8–14.3	0.04 *	11 (9.2)	4.0–14.5	0.77
Obesity	37 (8.7)	6.0–11.5	33 (8.9)	6.0–11.8	0.77	8 (6.7)	2.2–11.3	0.36
Thyroid disorder	27 (6.4)	4.0–8.7	25 (6.7)	4.2–9.3	0.42	13 (10.9)	5.2–16.6	0.02 *
Hyperlipidemia	23 (5.4)	3.3–7.6	23 (6.2)	3.7–8.7	0.07	5 (4.2)	0.5–7.9	0.48
Anxiety	23 (5.4)	3.3–7.6	18 (4.9)	2.7–7.0	0.16	6 (5.0)	1.1–9.0	0.82
Epilepsy	20 (4.7)	2.7–6.8	18 (4.9)	2.7–7.0	0.75	6 (5.0)	1.1–9.0	0.85
Arthritis	14 (3.3)	1.6–5.0	11 (3.0)	1.2–4.7	0.29	5 (4.2)	0.5–7.9	0.52
Recurrent urinary infections	9 (2.1)	0.7–3.5	9 (2.4)	0.9–4.0	0.26	1 (0.8)	0.0–2.5	0.25
Depression	8 (1.9)	0.6–3.2	6 (1.6)	0.3–3.0	0.27	0 (0)	0	0.07
Stroke	4 (0.9)	0.0–1.9	4 (1.1)	0.0–2.1	0.45	2 (1.7)	0.0–4.0	0.33
Hypertension	3 (0.7)	0.0–1.5	2 (0.5)	0.0–1.3	0.27	0 (0)	0	0.28
Cirrhosis	3 (0.7)	0.0–1.5	2 (0.5)	0.0–1.3	0.27	0 (0)	0	0.28
Valvular disease	3 (0.7)	0.0–1.5	1 (0.3)	0.0–0.8	<0.01 *	0 (0)	0	0.28
Neoplasia	2 (0.5)	0.0–1.1	2 (0.5)	0.0–1.3	0.60	1 (0.8)	0.0–2.5	0.49
Paralysis	2 (0.5)	0.0–1.1	2 (0.5)	0.0–1.3	0.60	1 (0.8)	0.0–2.5	0.02 *

* *p*-value < 0.05. ADHD: attention deficit hyperactivity disorder; CI: Confidence Interval; HC: hospital care; PC: primary care.

**Table 3 jpm-14-00956-t003:** Total use of PC and/or HC services by pediatric patients with chronic diseases.

Variables	Total 423 (100)	95% CI	PC Users 371 (87.7)	95% CI	*p*	HC Users 119 (28.1)	95% CI	*p*
*n* ≥ 1 (%)	386 (91.3)	88.6–94.0	371 (100)	100	<0.01 *	119 (100)	100	0.11
M (SD)	9.1 (8.4)	8.3–10.0	10.3 (8.3)	9.5–11.2		10.5 (7.8)	9.1–11.9	

* *p*-value < 0.05. CI: Confidence Interval; HC: hospital care; M (SD): mean (standard deviation); PC: primary care; *n* ≥ 1: number of patients with at least 1 contact.

**Table 4 jpm-14-00956-t004:** Use of PC services by pediatric patients with chronic diseases.

Variables	Total 423 (100)	95% CI	PC Users 371 (87.7)	95% CI	*p*	HC Users 119 (28.1)	95% CI	*p*
Total contacts								
*n* ≥ 1 (%)	371 (87.7)	84.6–90.9	371 (100)	100	<0.01 *	104 (87.4)	81.3–93.5	0.75
M (SD)	8.3 (8.4)	7.5–9.1	9.5 (8.3)	8.6–10.3		7.6 (7.6)	6.2–9.0	
Contact type								
Health-related								
*n* ≥ 1 (%)	371 (87.7)	84.6–90.9	371 (100)	100	<0.01 *	104 (87.4)	81.3–93.5	0.86
M (SD)	7.9 (8.0)	7.1–8.6	9.0 (7.9)	8.2–9.8		7.0 (7.0)	5.8–8.3	
Laboratory								
*n* ≥ 1 (%)	61 (14.4)	11.1–17.8	61 (16.4)	12.7–20.2	0.04 *	17 (14.3)	7.9–20.7	0.21
M (SD)	0.2 (0.6)	0.2–0.3	0.3 (0.7)	0.2–0.3		0.3 (0.7)	0.1–0.4	
Administrative								
*n* ≥ 1 (%)	23 (5.4)	3.3–7.6	23 (6.2)	3.7–8.7	0.85	7 (5.9)	1.6–10.2	0.70
M (SD)	0.2 (0.9)	0.1–0.3	0.2 (0.9)	0.1–0.3		0.2 (1.2)	0.0–0.5	
Contact form								
Face-to-face								
*n* ≥ 1 (%)	370 (87.5)	84.3–90.6	370 (99.7)	99.2–100	<0.01 *	103 (86.6)	80.3–92.8	0.81
M (SD)	8.2 (8.1)	7.4–8.9	9.3 (8.0)	8.5–10.1		7.4 (7.5)	6.0–8.8	
Telephone								
*n* ≥ 1 (%)	37 (8.7)	6.0–11.5	37 (10.0)	6.9–13.0	0.22	15 (12.6)	6.6–18.7	0.05
M (SD)	0.1 (0.7)	0.1–0.2	0.2 (0.8)	0.1–0.2		0.2 (0.5)	0.1–0.2	
Home								
*n* ≥ 1 (%)	2 (0.5)	0.0–1.1	2 (0.5)	0.0–1.2	0.60	1 (0.8)	0.0–2.5	0.49 *
M (SD)	0.00 (0.1)	0.0–0.0	0.01 (0.1)	0.00–0.01		0.01 (0.1)	0.0–0.03	
Professional contacted								
Pediatrician								
*n* ≥ 1 (%)	305 (72.1)	67.8–76.4	305 (82.2)	78.3–86.1	<0.01 *	82 (68.9)	60.5–77.4	0.78
M (SD)	4.9 (6.3)	4.3–5.5	5.6 (6.5)	4.9–6.2		4.3 (5.2)	3.3–5.2	
Nurse								
*n* ≥ 1 (%)	287 (67.8)	63.4–72.3	287 (77.4)	73.1–81.6	<0.01 *	78 (65.5)	56.9–74.2	0.54
M (SD)	1.9 (2.9)	1.6–2.2	2.2 (3.0)	1.9–2.5		1.7 (2.6)	1.2–2.2	
Family Physician								
*n* ≥ 1 (%)	81 (19.1)	15.4–22.9	81 (21.8)	17.6–26.1	0.30	22 (18.5)	11.4–25.6	0.66
M (SD)	0.7 (1.9)	0.5–0.9	0.8 (2.0)	0.6–1.0		0.6 (1.9)	0.3–1.0	

* *p*-value < 0.05. CI: Confidence Interval; PC: primary care; M (SD): mean (standard deviation); *n* ≥ 1: number of patients with at least 1 contact.

**Table 5 jpm-14-00956-t005:** Use of HC services by pediatric patients with chronic diseases.

Variables	Total 423 (100)	95% CI	PC Users 371 (87.7)	95% CI	*p*	HC Users 119 (28.1)	95% CI	*p*
Total contacts								
*n* ≥ 1 (%)	119 (28.1)	23.8–32.4	104 (28.0)	23.4–32.6	1.00	119 (100)	100	<0.01 *
M (SD)	0.8 (2.0)	0.6–1.0	0.9 (2.1)	0.6–1.1		3.0 (2.8)	2.5–3.5	
External consultations								
*n* ≥ 1 (%)	118 (27.9)	23.6–32.2	103 (27.8)	23.2–32.3	1.00	118 (99.2)	97.5–100	<0.01 *
M (SD)	0.8 (1.8)	0.6–1.0	0.8 (1.9)	0.6–1.0		2.9 (2.8)	2.4–3.3	
Hospitalizations								
*n* ≥ 1 (%)	5 (1.2)	0.2–2.2	5 (1.3)	0.2–2.5	0.70	5 (4.2)	0.5–7.9	<0.01 *
M (SD)	0.01 (0.1)	0.0–0.03	0.02 (0.1)	0.0–0.03		0.05 (0.3)	0.0–0.1	
Hospital day visits								
*n* ≥ 1 (%)	2 (0.5)	0.0–1.1	2 (0.5)	0.0–1.3	0.87	2 (1.7)	0.0–4.0	0.07
M (SD)	0.02 (0.3)	0.0–0.04	0.02 (0.3)	0.0–0.05		0.06 (0.6)	0.0–0.2	

* *p*-value < 0.05. CI: Confidence Interval; HC: hospital care; PC: primary care; M (SD): mean (standard deviation); *n* ≥ 1: number of patients with at least 1 contact.

**Table 6 jpm-14-00956-t006:** Multivariate analysis of the variables related to the use of PC and HC services.

Variables	Coefficient	Standard Error	Z Value	95% CI	*p* Value > |Z|
Primary Care service users					
Stroke	0.92	0.13	6.83	0.65–1.18	<0.01 *
Epilepsy	0.62	0.09	7.20	0.45–0.79	<0.01 *
Recurrent urinary infections	0.59	0.10	0.79	0.39–0.80	<0.01 *
Pneumonia	0.52	0.05	9.76	0.41–0.62	<0.01 *
Asthma	0.50	0.05	9.68	0.40–0.60	<0.01 *
Hyperlipidemia	0.35	0.10	9.68	0.40–0.60	<0.01 *
ADHD	0.34	0.08	3.65	0.16–0.53	<0.01 *
Thyroid disorder	0.31	0.09	3.55	0.14–0.48	<0.01 *
Anxiety	0.29	0.10	2.88	0.09–0.49	<0.01 *
European origin	0.21	0.07	3.07	0.08–0.34	<0.01 *
Female sex	0.15	0.034	4.41	0.08–0.22	<0.01 *
Complexity according to AMG	0.01	0.00	26.98	0.01–0.01	<0.01 *
Age	−0.20	0.02	−10.26	(−0.24)–(−0.16)	<0.01 *
Arthritis	−0.30	0.12	−2.45	(−0.53)–(−0.06)	0.01 *
Hospital Care service users					
Thyroid disorder	0.48	0.24	1.97	0.00–0.95	0.04 *
Hyperlipidemia	−0.96	0.39	−2.44	(−1.73)–(−0.19)	0.02 *
Recurrent urinary infections	−2.02	1.00	−2.02	(−3.98)–(−0.06)	0.04 *

* *p*-value < 0.05. ADHD: attention deficit hyperactivity disorder; AMG: Adjusted Morbidity Groups; CI: Confidence Interval.

## Data Availability

The data presented in this study are available on request from the corresponding author. The data are not publicly available due to their origin from the primary care electronic medical records of the Community of Madrid.

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
