# Peer review of "Utilization of Healthcare Services in Patients with Chronic Diseases under 18 Years Old: Differences and Contributing Factors"

_jpm, 2024, doi:10.3390/jpm14090956_

Round 1

Reviewer 1 Report

Comments and Suggestions for Authors

The manuscript is very interesting and I just have a few comments  that, I hope, will improve it:

- l.20: why authors didn't consider factors like income, material deprivation of the residence area, education from parents and geographic accessibility to the services. There is plenty literature about the relevance of these

- l.23: some literature refer to the designation super users. I suggest to use this as well as the cut-off for a user to be considered super

- l.26: did you consider the day and time of the utilization?

- l.45: indicate from where these rates are (e.g., country, city, social group) and give further information regarding differences on prevalence

- l.81: the introduction is too short and lacking of information about what literature says about, e.g., the topic, the reasons behind the patterns of pediatric utilization, what influences access, why the chronic illnesses require so many visits

- l.95: provide more information about the area and a map to help understanding the dynamics of the study area

- l.103: the database considered both public and private healthcare utilization?

- l.124: why not considering visits to the urgency?

- l.126: clarify which variables are quantitative and qualitative and how the different databases were linked and the unit of each of the variables

- l.138: the explanation of the process must be improved. it is not sufficient to indicate methods without explaining to which data has been applied and why

- l.147: include * on the table when p-value is significant. the same for the following tables

- l.149: is it possible that a user go to private hospitals or doctors and this utilization of the healthcare services is not considered here?

- l. 187: there should be more utilization intervals on the table to better understand the results

- l. 220: the results section is mostly about averages while it should be about the results that are statistically significant

- l. 255: the discussion is very short. it is missing more about what decision makers can do with this result, what should be done to improve access to healthcare and decrease the number of visits

- l.365: I do not agree that these results can be extrapolated for the entire country because not enough variables where consider and the authors cannot conclude that access to healthcare in madrid it is the same than in a rural area on the south of spain

Reviewer 2 Report

Comments and Suggestions for Authors

Dear Authors, thank you for your work. 

I have some questions for you: 

1) what is the role of digital health (telemedicine, etc) in Utilization of Healthcare Services in Patients with Chronic Diseases Under 18 Years Old?

2) did you analyze this subject with a gender perpective? I find a few information only in table 1.

3) what is COVID-19/SARS CoV 2 impact in this framework?

4) what is the vaccines importance in your data?

Round 2

Reviewer 2 Report

Comments and Suggestions for Authors

Great job!